# Fatty Acids Composition of Stomach Oil of Scopoli’s Shearwater (*Calonectris diomedea*) from Linosa’s Colony

**DOI:** 10.3390/ani12091069

**Published:** 2022-04-20

**Authors:** Francesco Giuseppe Galluzzo, Valentina Cumbo, Gaetano Cammilleri, Vittorio Calabrese, Andrea Pulvirenti, Nicola Cicero, Licia Pantano, Antonietta Mascetti, Giovanni Lo Cascio, Emanuela Bacchi, Andrea Macaluso, Antonio Vella, Salvatore Seminara, Vincenzo Ferrantelli

**Affiliations:** 1Dipartimento Alimenti, Istituto Zooprofilattico Sperimentale della Sicilia, Via Gino Marinuzzi 3, 90129 Palermo, Italy; francescogiuseppe92@gmail.com (F.G.G.); valentinacumbo@gmail.com (V.C.); licia.pantano@izssicilia.it (L.P.); giovanni.locascio@izssicilia.it (G.L.C.); bacchiemanuela@gmail.com (E.B.); andrea.macaluso@izssicilia.it (A.M.); laboratorio.residui@gmail.com (A.V.); salvatore.seminara@izssicilia.it (S.S.); vincenzo.ferrantelli@izssicilia.it (V.F.); 2Dipartimento di Scienze della Vita, Università degli Studi di Modena e Reggio Emilia, Via Università 4, 41121 Modena, Italy; andrea.pulvirenti@unimore.it; 3Dipartimento di Scienze Biomediche e Biotecnologiche, Università degli Studi di Catania, Torre Biologica Via Santa Sofia, 95123 Catania, Italy; calabres@unict.it; 4Dipartimento di Scienze Veterinarie, Università Degli Studi di Messina, Piazza Pugliatti 1, 98122 Messina, Italy; ncicero@unime.it (N.C.); zaghetta76@hotmail.it (A.M.)

**Keywords:** seabird, nutritional composition, stomach oil

## Abstract

**Simple Summary:**

Some Procellariiformes use an alternation of short tips and long trips to feed their chick. During these trips, they can accumulate stomach oil in the proventriculus derived from partially digested preys. The stomach oil was collected from adults of *Calonectris diomedea* from Linosa islands for fatty acids composition investigation. The results showed differences in fatty acid composition between the initial rearing period and the period near fledging. The present work gives a contribution to deepen the ecology and feeding strategies of the *C. diomedea* colony in Linosa island (Southern Italy).

**Abstract:**

*Calonectris diomedea* is a Procellariforms seabird having a very representative colony in Linosa Island (Southern Italy). The adult forms of *C. diomedea* produce a pasty oil from their proventriculus to feed their chicks during the rearing period. In this work, we examined the fatty acids composition of the stomach oil of *C. diomedea* from Linosa Island by gas chromatography with flame ionization detection (GC-FID). The samples were collected at 20 and 70 days after hatching. Twenty different fatty acids (FAs) were identified. Saturated fatty acids (SFA) were the most abundant in percentage (41.6%) at day 20 followed by polyunsaturated fatty acids (PUFA, 38.7%) and monounsaturated fatty acids (MUFA, 19.7%). MUFAs were the most abundant in samples collected at day 70 (53.8%), followed by SFAs (36.6%) and PUFAs (9.8%). Oleic acid (C18:1ω9) in the samples on day 70 was 4 times higher than that in the samples on day 20. The Principal Component Analysis (PCA) verified a clear separation of the stomach oil samples in two groups, according to the day of sampling. The results obtained confirm the role of FAs analysis of stomach oil to understand the ecology and breeding behaviour of *C. diomedea*, highlighting a resemblance with signatures recorded in marine organisms of Linosa Island.

## 1. Introduction

Scopoli’s shearwaters (*Calonectris diomedea*) is a long-distance migrant and colonial procellariform breeding on the northeast Atlantic and Mediterranean islands.

Scopoli’s shearwaters and other Procellariiformes (albatrosses, shearwaters, and storm petrels) have the peculiarity of visiting the colony infrequently, presumably because they forage over vast oceanic areas where the resources are believed to be scarce and unpredictable in location [1]. Their chicks accumulate large amounts of fat during nesting to deal with prolonged periods without feeding [2,3,4,5]. Partners use a flexible approach for foraging, including long and short trips [6]. Their diet includes predominantly epipelagic and mesopelagic squid and fish and crustaceans [7].

Most of the Procellariiformes feed their chicks with an oily paste as result of partial digestion of the preys in the proventriculus [4,8,9,10,11]. The chemical composition of stomach oil includes hydrocarbons, monoester waxes, diacylglycerol ethers, triacylglycerols, monoacylglycerols, cholesterols, alcohols and free fatty acids [9,10,12].

In procellarids, the nestlings accumulate a large lipid reserve to compensate for prolonged periods of parents’ absence [12]. The fat accumulation during the nestling stage is complemented with changes in FAs metabolism. In this period, considered to be energetically in need, a relatively rapid alteration in FA signatures might occur. Conversely, adults deplete their lipid stores because of the physiological stress associated with the programmed loss in response to physiological demands [13].

The analysis of the stomach oil in procellariids is useful to deepen the trophic relationship of adult forms and their breeding ecology [3,4,14]. A problem inherent in the fatty acid technique is that predator diets usually contain more than one prey species, such that the signatures are often complex and cannot be examined just by eye. The improvement in analytical chemistry procedures based on gas chromatography could be useful to overcome this concern.

Linosa Island hosts the second largest colony of Scopoli’s shearwater of the species [15], and a limited number of studies have assessed the provisioning strategies of Scopoli’s shearwater of Linosa’s colony, which are focused on GPS tracking and animal-borne video cameras [16,17]. This work aimed to study the FA content of the stomach oil collected from Scopoli’s shearwater chicks of Linosa Island in order to deepen their ecology and trophic relationship and highlight the possible role of fatty acids analysis as dietary tracers in *C. diomedea* trophic ecology.

## 2. Materials and Methods

### 2.1. Sampling Plan

The sampling plan was carried out in 2017 on Linosa island (Figure 1, Southern Italy, 35°52′30.2″ N 12°52′13.5″ E, Lat. 35.875056, Long. 12.870417) using the monitoring and sample collection protocols reported before [14]. The stomach oil was collected from chicks’ crops at 20 and 70 days after hatching [3] (Figure 2). Samples were placed into polypropylene tubes and stored at −20 °C until the analysis.

### 2.2. Fatty Acids Analysis

All the reagents used (n-hexane, potassium hydroxide, methanol) were HPLC grade (≥99.0%) and were purchased by Merck KGaA (Darmstadt, Germany). Individually FAs standards (GC purity ≥ 99.0%) of C14:0, C16:0, C17:0, C18:0, C20:0, C22:0, C16:1ω9, C17:1, C18:1ω9, C20:1ω9,C22:1ω9, C18:2ω6, C18:3ω3, C20:2ω6, C20:4ω3, C20:5ω3, C22:2ω6, C22:5ω3, C22:6ω3 were purchased from Merck KGaA (Darmstadt, Germany).

All the gases used for gas chromatography (GC) analysis were pure (≥99.9995%). Ultrapure water was obtained by a Milli-Q^®^ Integral 5 system (Merck KGaA, Darmstadt, Germany).

The FAs extraction was carried out as follows: 0.1 ± 0.001 g of the stomach oil sample was placed in a 5 mL test tube, then added with 0.2 mL of n-hexane and 0.2 mL of methanolic solution of potassium hydroxide 2N and vortexed for 1 min. The tubes were closed with the cap fitted with a tight polytetrafluoroethylene (PTFE) joint and shaken vigorously for 30 s. The solution was left until the upper n-hexane phase became transparent. This extract was transferred into a 1 mL vial for the GC-FID analysis. Fatty acid standards at 10,000 mg/L were prepared by adding 100 mg of the pure standard to a volumetric flask and diluted to 10 mL with n-hexane. A standard solution mixture at 500 mg/L was prepared by diluting 0.1 mL of each FAs standard soltion to 2 mL with n-hexane. The GC-FID analysis were carried out by a Trace GC/ULTRA HP 5890 GC + 7673 A/S chromatographer (Thermo Fisher, Waltham, MA, USA) with the conditions described by Galluzzo et al [18].

The injector port and the detector temperatures were 220 °C and 230 °C, respectively. The split ratio was 1:20. Compressed air and hydrogen flow rates were 350 mL min^−1^ and 35 mL min^−1^, respectively. The carrier gas was helium (1.5 mL min^−1^). The oven temperature was programmed at 6.0 °C min^−1^ from 130 to 225 °C, held for 15 seconds. A Famewax column (30 m × 0.25 mm i.d. × 0.25 μm df) was used for the separation. The qualification and quantification of the analytes were carried out using the ChromQuest 4.2.1 software. (Thermo Fisher, Waltham, MA, USA). The standard fatty acid mixture was used for the identification of the peaks. The FAs were identified by comparing their retention time and the peak positions by the formula:TR = TRst ± 0.5(1)
where TR is the retention time determined (min), and TRst is the retention time for each FAs standard. The relative percentages of the FAs were also determined. The quantitation of individual FA is based on comparing their peak areas (Ai) and the total area found in sample (∑A). The formula used to determine the relative percentages of fatty acids (C) was:Ai =A × 100/∑A(2)

Ten experimental measurements were conducted to test the precision. Standard deviation and repeatability were tested according to Taverniers et al. [19]. Peaks were adjusted and integrated manually when needed. Results were expressed as percentages (%).

### 2.3. Statistical Analysis

The samples were named alphabetically (A–H) and grouped by the period of sampling (day 20 and day 70 after hatching). A Principal Component Analysis (PCA) was carried out using the R software (4.1.2) (R Core Team). FA data were pre-treated with Pareto-Scaling [14,18,20]. The assumption of inter-correlation between variables was tested by the “cor” function of the R 4.0.2. software (R Core Team). The results of the test verified a correlation above 0.3 for all the variables considered. A Bartlett’s test was carried out to verify if the correlation matrix resembles an identity matrix. The results of the Bartlett’s test showed that the correlation matrix is significantly different from an identity matrix (Chisq = 1482.1; *p* value = <0.001). A Kaiser –Meyer–Olkin test was carried out to measure the sampling adequacy; the results of the KMO test showed an overall MSA of 0.5. Three different tests were conducted to assess the correct number of principal components (PCs) to keep: Kaiser–Harris criterion, Cattell Scree test and parallel analysis [21]. Moreover, 2 PCs explain the 99.2% of the variance and therefore were retained for the analysis. A Mann–Whitney U test was carried out to verify possible differences in FA contents between samples of day 20 and day 70 [22]. All the statistical analyses were carried out using the R 4.0.2. software (R Core Team).

## 3. Results

The FA contents of the samples sorted by sampling period are shown in Table 1 and Table 2. In total, 10 FAs were found in all 16 of the samples examined: 3 SFA (C16:0, C17:0, C18:0), 3 MUFA (C16:1, C17:1, C18:1ɷ:9) and 4 PUFA (C18:2ɷ:6, C20:4, C20:5, C22:6). C22:0, C22:1, C22:2 were found only in samples of day 20. The stomach oil samples collected on day 20 showed higher SFA amounts, accounting for 43.9% of the total FAs contents. C16:0 was the most representative SFA with percentage values between 24.3% and 26.8%, followed by C18:0. MUFA represented 19.7% of the total with a profile at day 20 that consisted of C18:1ω9 (10.6–15.6%), C16:1 (3.7–5.5%), C20:1 (0–2.6%), C17:1 (0.3–0.8%), C22:1 (0.3–0.7%). PUFA accounted for up to 41% of the total FAs contents. Docosahexaenoic acid (C22:6) was the most abundant in all the samples examined (22.1–26.1%), followed by eicosapentaenoic acid (C20:5; 8.2–8.6%) and linoleic acid (C18:2ω6; 1.6–2.1%). The PUFA contents at day 20 showed a ω3/ω6 ratio of 15.08.

The samples of day 70 revealed a significant increase in MUFA contents and a decrease in PUFA. In particular, the oleic acid (C18:1ω9) contents increased up to 4 times more than day 20, with a range between 45.1 and 56.8%, becoming the most representative fatty acid of the samples of day 70. Among the PUFA group, docosahexaenoic acid showed the highest decrease, ranging between 2.7 and 6.0%, followed by docosapentaenoic acid (C22:5). No docosadienoic acid (C22:2) was found in the samples of day 70. The ω3/ω6 ratio was lower (9.4) than day 20. The SFA contents were lower than samples of day 20, showing a decrease of myristic acid (C14:0) and arachidic acid (C20:0). Palmitic acid (C16:0) remained the most abundant SFA, accounting for 25.7% of the SFA amounts.

The first Principal Component (PC1) accounts for 97.8% of the total variance and PC2 accounts for 1.5%. The biplot is shown in Figure 3. C18:1ω9 performs the most significant contribution to both PCs, followed by C22:6ω3 and C20:5ω3.

A clear separation of the stomach oil samples into two groups was related to the collection time (day 20 vs. day 70). Samples collected at day 20 were characterized by a strongly negative coordinate on PC1, whereas samples collected at day 70 were correlated positively. This difference is due to the different content of FAs, which showed a heavy negative impact on PC1, such as C20:5ω3, C22:6ω3, C14:0, C20:0, C22:5ω3. Samples collected at day 20 share high content of these FAs C17:0, C22:5ω3, C14:0 and C22:6ω3 and low values of C18:1ω9, C17:1, C20:4ω3 and C18:0 that have a positive correlation with PC1. On the contrary, FAs profile in samples collected on the 70th day is characterized by high values of C18:0, C20:4ω3, and C18:1ω9.

Y axis position is influenced by C16:0 and C18:0 that are strongly negatively correlated with PC2 and C20:0 and C16:1 that are correlated positively. PC2 explains differences in FAs profiles within the same group. FAs profiles of samples collected on day 70 are more heterogeneous than samples collected on the 20th day. The individuals 70B, 70C, 70H, share high values of C18:0, C20:4ω3, and C16:0 (sorted from the strongest) and low values for the variables C16:1 and C14:0. 

The group in which the individuals 70D and 70E stand (positive coordinate on both the axis) share high values of C20:4ω3, C22:1 and C18:1ω9 and low values of C16:0, C22:6ω3, C22:5ω3, C20:0, C16:0, C20:1 and C20:5ω3.

The Mann–Whitney U test showed that the distributions of each FAs are different for the two groups and, therefore, were not from the same population (*p*-value < 0.05).

## 4. Discussions

Adult Procellariiformes generally feed their chicks large meals in intervals of more than one day [11,23,24]. The laying of a single egg and the prolonged nestling period resulting from slow growth and significant accumulation of fat are the most-cited adaptations to such an unpredictable environment [25,26,27]. Among Procellariiformes, Cory’s shearwater is characterized by prolonged incubation (54 days) and chick-rearing periods (90 days) [28].

Stomach oil is derived from the mechanical rupture of the prey in the proventriculus [3,4,29]. The structure of the neutral lipids accumulated in the stomach oil is linked to the lipid composition of the prey ingested, without any possible modification related to assimilation processes [3,4,30].

The differences between the composition of stomach oil samples collected at day 20 and day 70 (Figure 4) suggest that Scopoli’s shearwaters of Linosa island adopt a targeted strategy for provisioning their chicks, probably associated with the different nutritional requirements during the development and/or breeders for replenishing their reserves [3,31,32].

On this basis, it is possible to distinguish two development stages characterized by marked differences in FAs composition. The stomach oil samples collected 20 days after hatching are characterized by a higher SFA and PUFA content. Among PUFA, eicosapentaenoic acid (20:5ω3, EPA) and docosahexaenoic acid (C22:6ω3, DHA) are not synthesized by avian metabolism. These FAs play an essential role in developing the central nervous system and tissues [33]. Furthermore, an adequate provision of essential PUFA is essential during early development since the brain and retina require DHA during critical differentiation periods [34]. It was proven that the feeding behavior variation in Procellariiform seabirds is reflected in interspecific retinal differences [35]; this confirms the physiological and ecological importance of polyunsaturated fatty acids during the early stages of development.

Conversely, the stomach oil samples collected 70 days after hatching are characterized by a higher amount of MUFA, with a significant increase in the oleic acid (C18:1ω9) content, representing up to 50% of the total FAs content. Most lipids in migrating birds are composed of unsaturated FAs, mainly C18:1ω9 and C18:2, which predominate over saturated FAs [36,37]. Three FAs (C18:1ω9, C18:2 and C16:0) usually comprise at least 75% of the FAs in body fat of birds during migration [37]. Therefore, it is assumed that the increase in oleic acid of the samples from day 70 is linked to a targeted supply of the chicks before the first flight (about 90 days) to increase physical activity during migration [37].

Among the SFAs, palmitic acid (C16:0) was the most abundant SFA found in the stomach oil samples examined. No statistical differences in the palmitic acid content were found between samples of days 20 and 70, probably related to its ecological importance throughout the rearing period. The abundance of this FA is considered to be important for the release of energy during migration and may explain why it is found in such high levels in fledgling shearwaters [38].

The results of this work could confirm that Procellariiformes adopted a dual provisioning strategy consisting of short-distance trips carried out mainly in the shoreface zone near the colonies and long-distance foraging trips offshore [6,39]. A similar trend was found in common guillemots of Scotland [40] and other seabirds [41,42], highlighting how the demands of chick rearing constrain this species to local foraging areas around breeding colonies, leading to possible seasonal changes in diet. Cory’s shearwaters show their highest food delivery rates during the first 29 days after hatching, suggesting a higher frequency of short trips for chicks’ provisioning, which appears to be more profitable [43,44]. 

This condition leads the parents to catch prey predominantly in the intertidal zone. This consideration could be supported by the fatty acids profile obtained in the stomach oil samples of day 20 showed a high probability of resemblance with signatures recorded in black mussels (*Mytilus galloprovincialis*) [45,46,47]. Furthermore, the color of stomach oil samples collected at day 20 range from deep orange to black due to the presence of valve fragments attributable to black mussels, suggesting a prevalent supply of this species during the first development stage of the chicks. Female herring gulls in the pre-laying period preferred mussels, which provided calcium for egg-shell formation [48].

The fatty acids analysis of the stomach oil samples at day 70 verified a significant increase in the MUFA contents, in particular oleic acid (18:1ω9), which accounting for 56% of the total. The fatty acids profile obtained in this work seems to exclude the provisioning of squid or other cephalopods for the Scopoli’s shearwater colony of Linosa island, in contrast to what was found in the Cory’s shearwater colony of the Azores [6,49,50]. The comparison of our data with the fatty acid profile of Mediterranean cephalopods reported in the literature showed a low match [51,52,53,54].

Unfortunately, the data available in the literature on the FA profiles of possible Mediterranean prey species did not allow us to find a reliable match with the results obtained for the samples of day 70. However, it should be noted that, to our knowledge, most studies about the fatty acid composition of Mediterranean organisms are referred to species of commercial interests [52,53,55,56,57,58], thus significantly limiting comparisons. In addition, it should also be considered that prey species might also exhibit spatial and temporal variability in lipid content or whole body FA signatures reflecting dietary shifts or changes in the food web base [12].

## 5. Conclusions

The results clearly showed that there are differences in FAs’ composition within the rearing period. FA signatures in stomach oil are undoubtedly influenced by the dietary FA intake, and we argue that they can provide valuable insights into the seabird diets.

This study confirms the role of FAs of stomach oils to investigate the trophic relationship of *C. diomedea* in Linosa island and the foraging strategies during breeding. 

## Figures and Tables

**Figure 1 animals-12-01069-f001:**
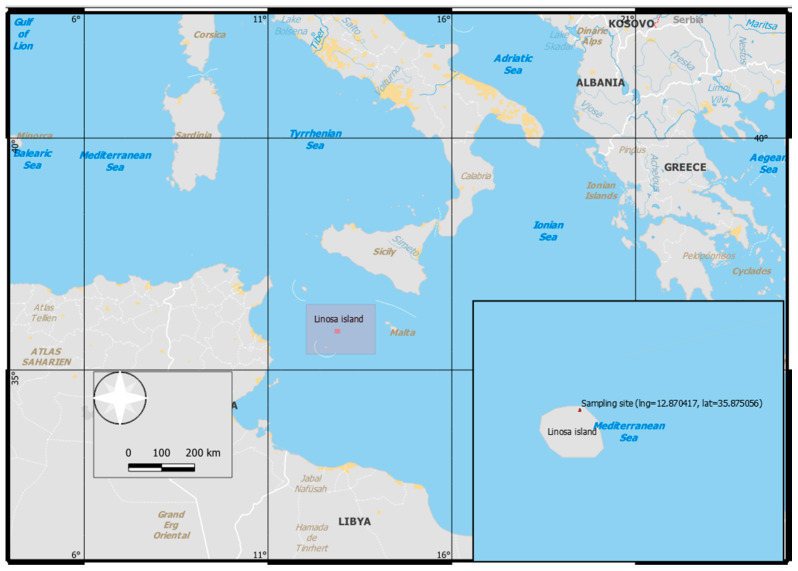
Location of Linosa Island (South Mediterranean; 35°52′30.2″ N 12°52′13.5″ E), with the study colony marked in red.

**Figure 2 animals-12-01069-f002:**
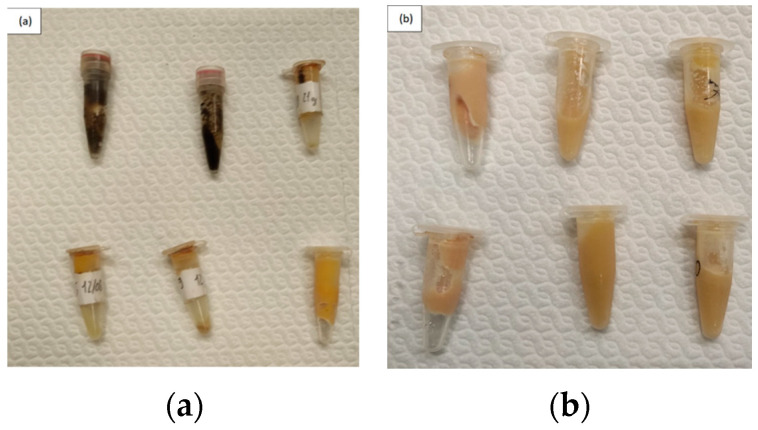
(**a**) Samples collected 20 days after the hatching; (**b**) Samples collected 70 days after the hatching.

**Figure 3 animals-12-01069-f003:**
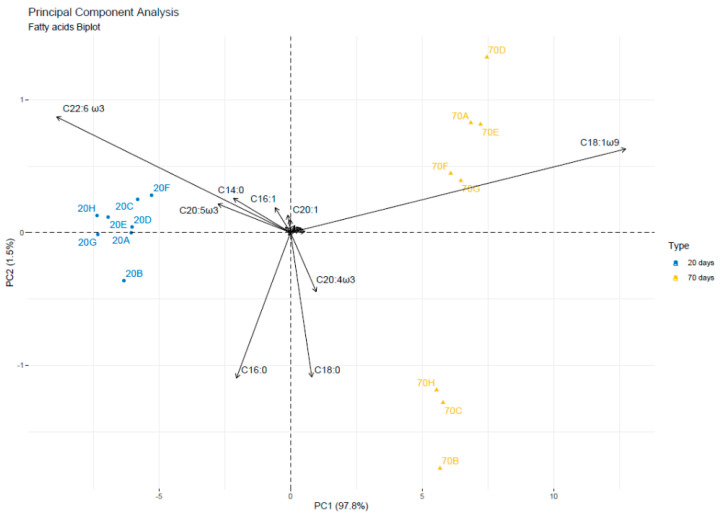
PC1 vs. PC2 biplot of FAs patterns of the stomach oil samples analyzed, according to the sampling period (day 20 vs. day 70).

**Figure 4 animals-12-01069-f004:**
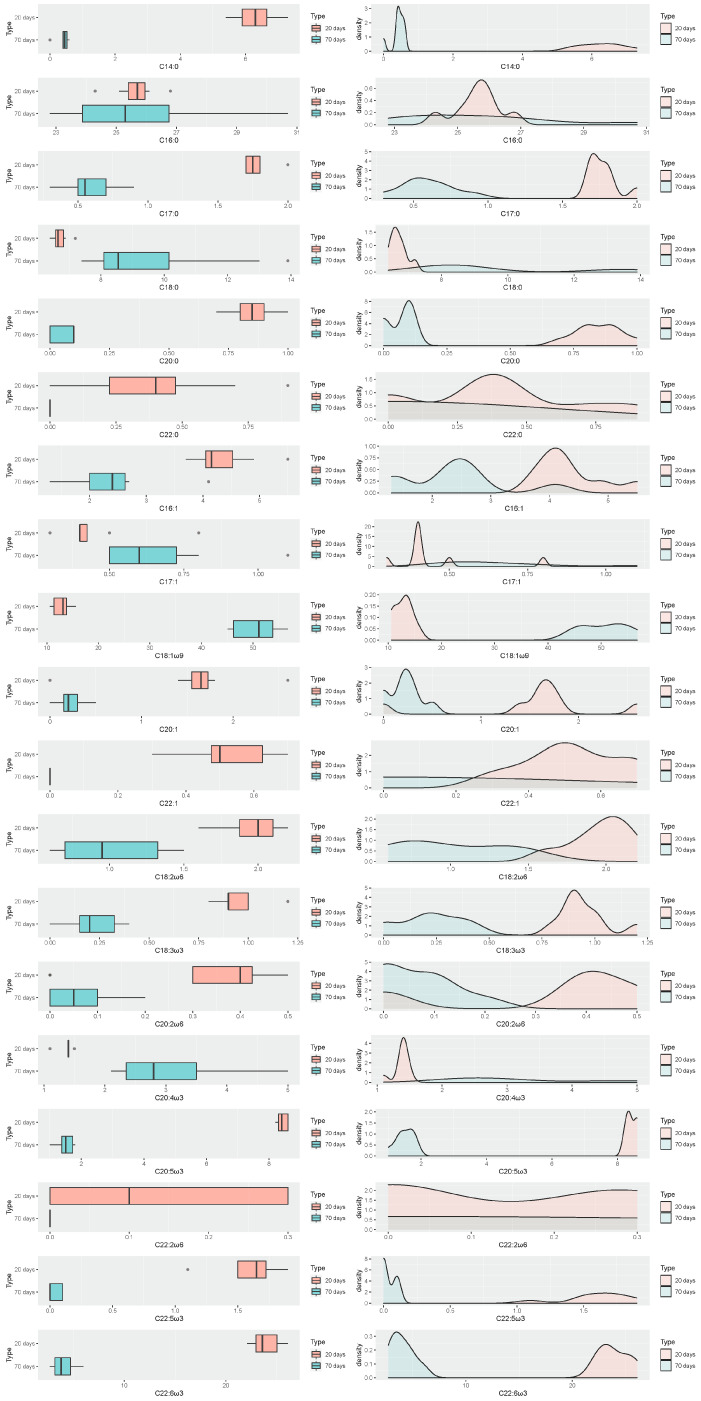
Boxplot and distribution plots of FAs according to the sampling period (day 20 vs. day 70).

**Table 1 animals-12-01069-t001:** Fatty acids composition of the stomach oil samples collected at day 20 (expressed as %). n.d. = not detected.

Fatty Acid	20A	20B	20C	20D	20E	20F	20G	20H	Mean ± SD
C14:0	5.4	6.6	7.3	5.6	6.5	6.0	6.8	6.1	6.29 ± 0.63
C16:0	25.8	26.8	24.3	25.9	25.6	25.1	26.1	25.5	25.64 ± 0.73
C17:0	1.7	2.0	1.7	1.7	1.8	1.7	1.8	1.8	1.78 ± 0.10
C18:0	6.4	7.2	6.8	6.7	6.6	6.5	6.6	6.9	6.71 ± 0.25
C20:0	0.8	0.9	0.7	0.9	0.8	0.8	1.0	0.9	0.85 ± 0.09
C22:0	0.4	0.4	0.3	0.4	n.d.	n.d.	0.9	0.7	0.52 ± 0.23
ΣSFA ^1^	40.5	43.9	41.1	41.2	41.3	40.1	43.2	41.9	41.65 ± 1.30
C16:1	4.9	4.2	5.5	4.1	4.1	4.4	3.9	3.7	4.35 ± 0.59
C17:1	0.8	0.3	0.4	0.4	0.4	0.5	0.4	0.4	0.45 ± 0.15
C18:1ω9	13.4	12.8	13.8	13.8	11.6	15.6	10.6	10.7	12.79 ± 1.73
C20:1	1.7	n.d.	2.6	1.8	1.6	1.6	1.4	1.7	1.77 ± 0.39
C22:1	0.5	0.5	0.7	0.4	0.6	0.7	0.3	0.5	0.53 ± 0.14
ΣMUFA ^2^	21.3	17.8	23.	20.5	18.3	22.8	16.6	17.	19.66 ± 2.57
C18:2ω6	1.8	2.2	1.6	2.0	2.1	1.9	2.1	2.0	1.96 ± 0.19
C18:3ω3	1.2	0.9	0.9	0.9	1.0	0.8	1.0	0.9	0.95 ± 0.12
C20:2ω6	0.5	n.d.	0.4	0.4	0.5	0.4	n.d.	0.4	0.43 ± 0.05
C20:4ω3	1.5	1.4	1.1	1.4	1.4	1.4	1.4	1.4	1.38 ± 0.12
C20:5ω3	8.6	8.3	8.4	8.2	8.6	8.3	8.6	8.4	8.43 ± 0.16
C22:2ω6	n.d.	0.2	0.3	n.d.	0.3	0.3	n.d.	n.d.	0.28 ± 0.05
C22:5ω3	1.5	1.8	1.1	1.7	1.6	1.5	1.7	1.9	1.60 ± 0.24
C22:6ω3	23.1	23.5	22.1	23.7	24.9	22.5	25.4	26.1	23.91 ± 1.42
ΣPUFA ^3^	38.2	38.3	35.9	38.3	40.4	37.1	40.2	41.1	38.69 ± 1.77
ratio ω3/ω6	15.61	14.96	14.61	14.96	12.93	13.27	18.14	16.13	15.08 ± 1.54

^1^ SFA: saturated fatty acids. ^2^ MUFA: monounsaturated fatty acids. ^3^ PUFA: polyunsaturated fatty acids.

**Table 2 animals-12-01069-t002:** FAs composition of the stomach oil samples collected at day 70 (expressed as %). n.d. = not detected.

Fatty Acid	70A	70B	70C	70D	70E	70F	70G	70H	Mean ± SD
C14:0	0.4	0.4	0.4	0.5	0.5	0.6	0.6	n.d.	0.5 ± 0.09
C16:0	23.9	27.2	26.6	22.8	23.8	24.8	25.8	30.7	25.7 ± 2.51
C17:0	0.3	0.7	0.7	0.5	0.6	0.5	0.5	0.9	0.59 ± 0.18
C18:0	8.3	13.9	13.	7.4	7.5	9.2	8.6	8.5	9.55 ± 2.5
C20:0	n.d.	0.1	0.1	n.d.	0.1	0.1	0.1	n.d.	0.1 ± 0
ΣSFA	32.9	42.3	40.8	31.2	32.5	35.2	35.6	40.0	36.31 ± 4.2
C16:1	2.5	1.3	1.4	2.6	2.7	2.3	2.2	4.1	2.39 ± 0.87
C17:1	0.5	0.5	0.5	0.7	0.8	0.6	0.6	1.1	0.66 ± 0.21
C18:1ω9	53.7	45.1	46.3	56.8	54.4	50.2	52.1	46.2	50.6 ± 4.36
C20:1	0.5	0.2	0.3	n.d.	0.2	0.3	0.2	n.d.	0.28 ± 0.12
ΣMUFA	57.2	47.1	48.5	60.1	58.1	53.4	55.1	51.4	53.85 ± 4.64
C18:2ω6	0.6	0.7	0.7	1.5	1.4	0.9	1	1.3	1.01 ± 0.35
C18:3ω3	0.2	0.2	0.2	n.d.	0.4	0.4	0.3	n.d.	0.28 ± 0.1
C20:2ω6	n.d.	0.1	0.1	n.d.	0.2	0.1	n.d.	n.d.	0.13 ± 0.05
C20:4ω3	2.8	5.0	4.4	2.1	3.2	2.2	2.4	2.8	3.11 ± 1.05
C20:5ω3	1.6	1.4	1.4	1.3	1.0	1.7	1.8	1.8	1.50 ± 0.28
C22:5ω3	n.d.	0.1	0.1	n.d.	n.d.	0.1	n.d.	n.d.	0.1 ± 0
C22:6ω3	4.7	3.1	3.80	3.8	3.2	6.0	4.8	2.7	4.01 ± 1.09
ΣPUFA	9.9	10.6	10.7	8.7	9.4	11.4	9.3	8.6	9.83 ± 1.01
ratio ω3/ω6	15.5	12.25	12.38	4.8	4.88	10.4	9.3	5.62	9.4 ± 3.98

## Data Availability

The data presented in this study are available on request from the corresponding author. The data are not publicly available due to privacy restrictions.

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
