# Peer review of "Fatty Acids Composition of Stomach Oil of Scopoli’s Shearwater (Calonectris diomedea) from Linosa’s Colony"

_animals, 2022, doi:10.3390/ani12091069_

Round 1

Reviewer 1 Report

This is a nice piece of seabird work on a subject that has not been investigated in the past. The study area is representative for central Mediterranean region and the colony size can provide an adequate sample for analysis. The authors have studied the fatty acids in the stomach of young Cory’s shearwaters at two different developmental stages and have found clear cult differences. The analytical procedure is robust and substantiate the aim of the study and its findings. I would suggest a number of changes that could improve the manuscript and make it more comprehensive.

  1. In the introduction section some more comments are needed on the significance of fatty acids and the nutritional and structural value for the species of the different roles in the development of the species nestlings. There are some data offered in the “Discussion” section, but they could be enriched and presented in the “Introduction” section.
  2. Figure 1. Should be enlarged and show the islet colony in the broader geographical section of central Mediterranean. By this way the reader could obtain an idea on the spatial range of foraging shearwaters.
  3. Fatty acids differ greatly by grouping the development periods of the young birds. However, it would be interesting if some data were presented on the nest or parent quality of the these young. If the colony is long monitored breeding success of this site tenacious species could also elucidate the reason for the observed differences apart the age of the chicks.
  4. In the “Discussion” section the nutritional properties of the most dominant fatty acids of each period (if available from the literature) would be useful.
  5. A question that arises by reading the “Results” and the “Discussion” sections is how parent birds differentially select prey species with diverse fatty acid content. Is this process behaviorally triggered or just adaptive? What about if short foraging sorties deplete the available food sources near the colony during the nestling stage and parent birds have to pursue long foraging trips as the chick-rearing stage progress and their food requirements increase. Some commenting on that would be interesting.

Author Response

Dear Reviewer

Please find attached the revised version of our paper. We have carefully revised our manuscript. We apologize for the delay but we have done our best to respond to your requests in the best possible way, adhering as closely as possible to the high journal's standards and aims. We have attached a word file with the track changes made to ease your perusal of our manuscript changes.

All the step by step changes are reported below:

  1. In the introduction section some more comments are needed on the significance of fatty acids and the nutritional and structural value for the species of the different roles in the development of the species nestlings. There are some data offered in the “Discussion” section, but they could be enriched and presented in the “Introduction” section.

Dear reviewer, thank you for your precious suggestions. We added more information on the role of lipids and FAs in the main document (see lines 52-58)

  1. Figure 1. Should be enlarged and show the islet colony in the broader geographical section of central Mediterranean. By this way the reader could obtain an idea on the spatial range of foraging shearwaters.

Dear reviewer, the figure was modified according to your precious suggestions.

  1. Fatty acids differ greatly by grouping the development periods of the young birds. However, it would be interesting if some data were presented on the nest or parent quality of the these young. If the colony is long monitored breeding success of this site tenacious species could also elucidate the reason for the observed differences apart the age of the chicks.

Dear reviewer, unfortunately, due to the principal aim of this work, we have not taken into consideration these aspects but we reserve the right to propose them for sure in next works.

  1. In the “Discussion” section the nutritional properties of the most dominant fatty acids of each period (if available from the literature) would be useful.

Dear reviewer, unfortunately, we encountered many difficulties to find other references on the role of the dominant fatty acids of each period in seabirds. Most of this information are reported in lines 214-249.

  1. A question that arises by reading the “Results” and the “Discussion” sections is how parent birds differentially select prey species with diverse fatty acid content. Is this process behaviorally triggered or just adaptive? What about if short foraging sorties deplete the available food sources near the colony during the nestling stage and parent birds have to pursue long foraging trips as the chick-rearing stage progress and their food requirements increase. Some commenting on that would be interesting.

Dear reviewer, many thanks for your comments. Given our results and what is reported in literature for other seabirds we have reason to think that the process is be behaviourally triggered. Other comments are added in lines 241-244 of the revised MS.

Reviewer 2 Report

The work submitted for review concerns Calonectris diomedea, a Celtic seabird with a representative colony on the island of Linosa (Italy), which produces gastric oil from its blood chambers with which it feeds its chicks during the young rearing period. The aim of this study was to evaluate the fatty acid profile of C. diomedea stomach oil by gas chromatography with flame-ionization detection (GC-FID). Suggestions and comments on the typescript are provided below:
Line 38: Introduction - in my opinion, if the aim of the paper was to evaluate the fatty acid profile in the stomach oil of the birds selected for observation then the introduction should include a brief characterization guided towards understanding the conclusion of the study after reading the paper, which was formulated: "The results obtained confirm the role of FAs analysis of stomach oil to understand the ecology and breeding behavior of C. diomedea highlighting a resemblance with signatures recorded in marine organisms of Linosa island" and "This study confirms the role of FAs of stomach oils to investigate the trophic relationship of C. diomedea in Linosa island and the foraging strategies during the breeding" 
Line 105: Was detector response (Response of the Detector-RD) included in the calculations?
Line 256: Reference - please check literature - minor errors.

Author Response

Dear Reviewer

Please find attached the revised version of our paper. We have carefully revised our manuscript. We apologize for the delay but we have done our best to respond to your requests in the best possible way, adhering as closely as possible to the high journal's standards and aims. We have attached a word file with the track changes made to ease your perusal of our manuscript changes.

All the step by step changes are reported below:

Line 38: Introduction - in my opinion, if the aim of the paper was to evaluate the fatty acid profile in the stomach oil of the birds selected for observation then the introduction should include a brief characterization guided towards understanding the conclusion of the study after reading the paper, which was formulated: "The results obtained confirm the role of FAs analysis of stomach oil to understand the ecology and breeding behavior of C. diomedea highlighting a resemblance with signatures recorded in marine organisms of Linosa island" and "This study confirms the role of FAs of stomach oils to investigate the trophic relationship of C. diomedea in Linosa island and the foraging strategies during the breeding" 
#R: Dear  reviewer, many thanks for this precious suggestion. We added this information in the main document.

Line 105: Was detector response (Response of the Detector-RD) included in the calculations?
#R: Dear reviewer, we confirm that the response of the detector was included in the calculations.

Line 256: Reference - please check literature - minor errors.

#R:We have checked the literature according to tour suggestions.

Hope these changes could be helpful for the manuscript reconsideration.
